# Elderberry Concentrate Juice Industrial By-Products Characterization and Valorisation

**Maria Inês Veloso [1], Elisabete Coelho [1,*], Oswaldo Trabulo [2] and Manuel A. Coimbra [1]**

[1] LAQV-REQUIMTE, Department of Chemistry, University of Aveiro, 3810-193 Aveiro, Portugal
[2] Indumape—Industrialização de Fruta, SA, Parque Industrial Manuel da Mota, Rua Bartolomeu Dias, 11-13, 3100-354 Pombal, Portugal
[*] Correspondence: ecoelho@ua.pt; Tel.: +351-234-247365

**Abstract:** The elderberry juice industry generates by-products that can be valorised as a source of valuable compounds, namely anthocyanins and carbohydrates recovered from pomace and retentate. This work aims to valorise the anthocyanins and carbohydrates present in pomace and retentate, focused on the analysis and characterization of the polysaccharides present and their use for the stability of pink beverages. The present work shows that pomace contains 50% of carbohydrates, in which glucose and xylose are the major constituents, probably arising from cellulose and xyloglucans of cell walls polysaccharides and from the moiety of main anthocyanins present in berry skin. The concentrated juice is rich mainly in free sugars (55%), glucose and fructose, containing also pectic polysaccharides. However, a large amount of compounds are retained in the ultrafiltration membranes of 100 kDa, constituting the retentate, comprising 52% of water insoluble material. The remaining 48% constitutes the water-soluble fraction, composed of 47% of free sugars, mainly fructose (80%), and 14% of polymeric material constituted by 38% of pectic polysaccharides and 44% of anthocyanins. The use of the colourant in the form of a complex pectic polysaccharides-anthocyanins allowed to achieve a higher colour stability than the isolated anthocyanins, over more than 22 days. This property allowed to use the retentate water soluble fraction as a natural colourant ingredient to develop a stable pink tonic water.

**Keywords:** elderberry; juice processing; retentate; pomace; carbohydrates; anthocyanins; colourant; tonic water; polysaccharides; colour stability

## 1. Introduction

Elderberry (*Sambucus nigra* L.) fruit has a dry matter that constitutes 22–29% of its weight, mainly composed of carbohydrates (18%) and anthocyanins (1%) [1–4]. Elderberries have been widely used throughout history as a folk medicine, mainly due to their anthocyanins, which are potent antioxidant agents and present anticarcinogenic, immuno-stimulating, antibacterial, antiallergic and antiviral properties, and, therefore, may contribute to the prevention of several degenerative diseases such as cardiovascular disease, cancer, inflammatory disease and diabetes [4,5]. Nowadays, elderberry is commercialized in form of dietary supplements [6,7]. The black-purple-red colour of this berry is due to the presence and concentration of anthocyanins, making it highly demanded as a natural colourant for the food industry [8–10]. The berry and its juice have been proposed to solve colour losses caused by food processing, as canning [11–15]. Its addition to wine to improve and emphasize colour, although considered an adulteration and an illegal practice [16], is still a traditional non-commercial practice highly appreciated and valued by local communities.

Elderberry fruit is very seasonal (1 month in August in the north of Portugal). The main cultivars are "Bastardeira", "Sabugueira" and "Sabugueiro", with similar anthocyanins profiles [1]. Only for cyanidin-3,5-diglucoside and cyanidin-3-sambubioside-5-glucoside

were there observed significant differences between the cultivars. The levels of "Bastardeira" are higher than those of "Sabugueiro". "Bastardeira" was reported as the best cultivar concerning total soluble solids, anthocyanins, polyphenols, and antioxidant activity, but containing a lower total free sugar content when compared to "Sabugueiro" [1]. Despite these differences, the producers commercialize them as blends. Due to its perishability, it is mostly transformed into juices, stored in a frozen state, or juice concentrate, to allow its storage under refrigeration conditions. The thermally processed juice is a safe alternative to the elderberry fruit consumption due to the presence of sambunigrin, a potentially toxic cyanogenic glycoside. As thermal processing significantly decreases the levels of this compound, it is safer to consume the berries as transformed products rather than fresh [1,8]. To obtain a concentrated juice, elderberry fruit is firstly destemmed and then mechanically pressed. The resulting must undergoes maceration with enzymes to increase the juice extraction efficiency, enhancing the release of anthocyanins [17]. A new enzymatic clarification/depectinisation followed by ultrafiltration is intended to reduce suspended substances, contributing to a final product with less turbidity. The last step is water evaporation and, afterwards, the obtained concentrate is homogenized and stored.

The production process of concentrated juice generates a large amount of by-products (25–40% of the total berry weight) [5,18]. The pomace, which is composed by the skin and smashed seeds, results from the pressing, while the retentate, which is a sludgy residue, results from the clarification of the juice through the ultrafiltration process [19,20]. At present, the pomace is used for animal feed, fertilization, or burned, while the retentate is discarded as industrial effluent. The large amount of waste produced in addition to the great loss of valuable compounds also raises serious management problems, both from the economic and environmental point of view [21,22]. Therefore, there is an urgent necessity for their valorisation. Research on elderberries has been focused on berries and flowers, and on their polyphenol content. In addition, elderberry polysaccharides, namely the xyloglucans, homogalacturonans, and rhamnogalacturonans, have been reported to have *in vitro* immunomodulatory activity [6,7]. Triterpenic acids, namely the ursolic and oleanolic acids, are the main components in the fruit lipophilic fraction [23]. Regarding to by-products exploration, pomace and stems have been valorised as a potential source of anthocyanins [18,24]. As far as we know, there is no available information on retentate characterization and valorisation. However, the retentate from other fruit concentrated juice, such as apple, was revealed to be a source of valuable compounds, enriched during the ultrafiltration process, such as protein and β-sitosterol, and successfully used as an ingredient for food and feed applications [20]. Thus, in the present work, the carbohydrate composition of pomace, retentate, as well as the concentrated juice obtained in the same process were studied as potential and valuable sources of carbohydrates and anthocyanins able to be incorporated as alternative natural colourant food ingredients.

## 2. Materials and Methods

### 2.1. Samples

The concentrated elderberry juice and its by-products, pomace and retentate, made from Portuguese elderberry cultivars ("Bastardeira", "Sabugueira" and "Sabugueiro") from Varosa Valley, were supplied by Indumape, S.A., Portugal. The industrial parameters of the ultrafiltration system (polyvinylidene fluoride—PVDF membrane with 100 kDa cut-off) used for juice clarification (retentate obtention) are the following: configuration with 37 tubules ($\frac{1}{2}$″ diameter) in polysulfone shell with a membrane area of 5.1 m$^2$, pressure between 1.5 and 5.5 bar (maximum operating pressure: 6.2 bar at 49 °C and minimum outlet pressure of 0.7 bar), and typical feed side pressure drop per module: 0.27–0.40 bar.

### 2.2. Samples Preparation

To separate the polymeric compounds from the free sugars, the concentrated juice was dialysed through a membrane with a 12 kDa cut-off, against water. The high molecular weight material was submitted to solid-phase extraction in C18 cartridges (SPE-C18,

Supelco-Discovery, 10 g). The column was preconditioned with methanol followed by water. The sample was loaded into the column, and the unbound material (sugars and salts, including unprotonated organic acids) was eluted with water. The retained hydrophobic fraction was eluted using acidic methanol (2% *v/v* acetic acid in methanol), followed with acidic acetone, if needed [25].

Pomace was milled, frozen, and freeze-dried before sugar analysis.

The retentate was fractionated and isolated based on the procedure described by Cruz et al. [20]. The suspension (150 g) was centrifuged (15,000 rpm, 4 °C, 20 min) to separate the water soluble from the insoluble compounds (Figure 1). The insoluble material was washed with 120 mL of water at 40 °C during 15 min. The retentate water soluble material was dialysed. The material that was diffused through the dialysis membrane (dialysate) was recovered by concentration under rotary-evaporation. The high molecular weight material (HMWM) was passed through a C18 column as described for the concentrated juice. All samples were frozen and then freeze-dried before analysis.

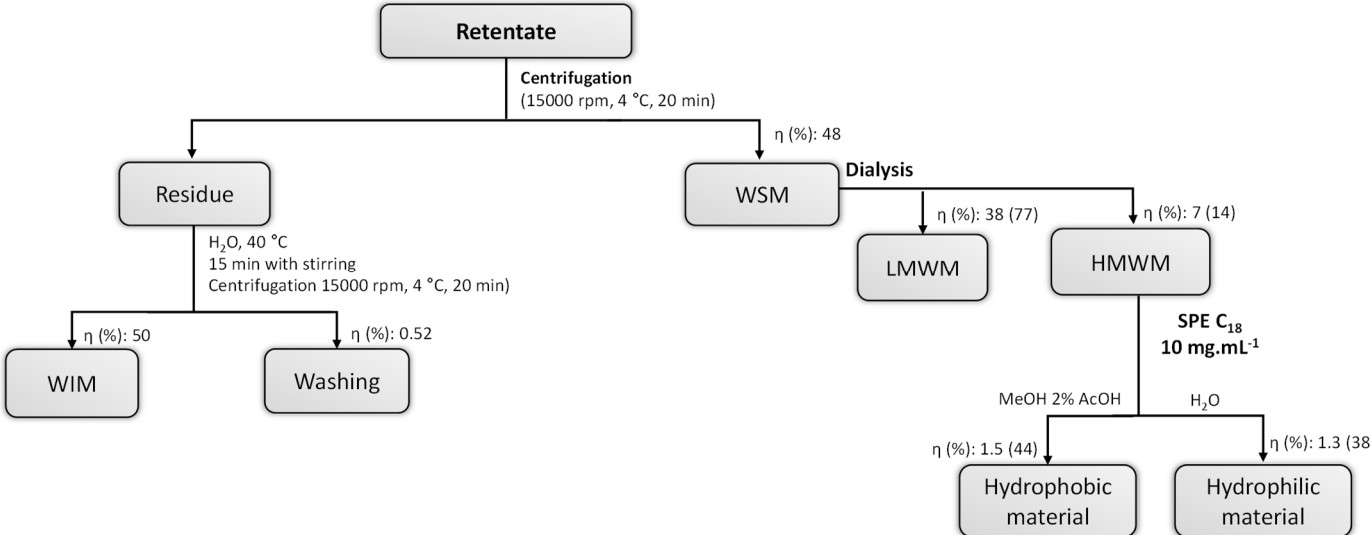

**Figure 1.** Schematic representation of the fractionation steps of the retentate. Yields are expressed on a dry weight basis (the values between brackets correspond to yields calculated based on the weight basis of water-soluble material, WSM, or on high molecular weight material, HMWM). AcOH—acetic acid. LMWM—low molecular weight material. MeOH—methanol. WIM—water insoluble material.

*2.3. Sugar Analysis*

Neutral sugars were analysed by gas chromatography-flame ionization detection (GC-FID) after conversion to their alditol acetates, using 2-deoxyglucose as the internal standard. Briefly, freeze dried samples were treated with 1 M $H_2SO_4$ during 2.5 h to release the monosaccharides, followed by neutralization, reduction with $NaBH_4$ and acetylation using acetic anhydride/methylimidazole. For the determination of free sugars composition, the same procedure was followed, except the hydrolysis step that was omitted. Fructose was quantified as the sum of mannitol and sorbitol due to the epimerization of fructose during the reduction step, using the ratio of the epimerization reaction to mannitol (43%) and glucitol (57%). The use of a DB-1 column instead of a DB-225 allowed the quantification of sucrose and other disaccharides, if present [25,26]. Total sugars were corrected considering the degradation of free sugars during the acid hydrolysis step [27]. Due to the high hygroscopicity of the concentrated juice was not possible to use the freeze-dried sample, the sample was dried by centrifugal evaporation until dryness and the results expressed on a dry weight basis.

Total uronic acids content was determined by *m*-phenylphenol colorimetric method with slightly modifications. In this case, the addition of the standard to the sample was

used instead of doing an external calibration curve. This was due to the samples red colour resultant from the high quantity of anthocyanins, which interfered with the pink colour of the analyte signal. Free uronic acids were also determined by GC-FID [28], allowing to distinguish the different types of uronic acids in samples with high uronic acids content. Briefly, samples were reduced with $NaBH_4$ in $NH_3$ 3 M (15% *m/v*), for 1 h, at 30 °C. The mixture was filtered and loaded onto a column Dowex 50WX8, the eluate was evaporated, added methanol, and dried again. The next step was a lactonization at 85 °C, during 4 h. Afterwards 200 μL pyridine and 400 μL n-propylamine were added [28], sequentially, kept at 55 °C, 30 min, and in the end evaporated until dryness. Pyridine (400 μL) was added and then an acetylation took place (1 mL of acetic anhydride, 95 °C, 1.5 h). The samples were injected (2 μL, split ratio of 1:20) in a GC-FID PerkinElmer-Clarus 400 with a capillary column DB-225 (30 m length, 0.25 mm inner diameter and 0.15 μm film thickness). The oven temperature program was as follows: 80 °C to 210 °C at a rate of 6 °C/min, and hold at this temperature during 6 min, increasing to 230 °C at a rate of 10 °C/min, holding 30 min at 230 °C. The uronic acids were identified and quantified based on their retention times and response factors obtained by injection of standards.

### 2.4. Glycosidic-Linkage Analysis

Glycosidic-linkages present in the SPE-C18 water fraction, obtained from the water soluble high molecular weight material of retentate were determined. After methylation in DMSO solution saturated with NaOH, a dialysis was performed against a solution of 50% aqueous ethanol. This procedure was repeated one more time.

To determine the glycosidic-linkages of uronic acids, the solutions were split into two portions and, in one of them, a carboxyl reduction was performed [29,30]. The permethylated polysaccharides were dried and dissolved in 1 mL of anhydrous tetrahydrofuran, 20 mg of lithium aluminium deuteride was added under argon and the suspension was kept at 65 °C during 4 h. The excess of reagent was destroyed with ethanol, and the pH of the mixture was neutralized with 1 M HCl. The reduced polymer was isolated by addition of 2 mL of $CHCl_3$:MeOH (2:1, *v/v*), the precipitate was removed by centrifugation and the supernatant was evaporated. Both fractions, the carboxyl-reduced material and the non-reduced one, were then converted into their partially methylated alditol acetates form. Firstly, they were hydrolysed with 0.5 mL of 2 M trifluoroacetic acid (1 h at 121 °C) and dried by centrifugal evaporation. The reduction of monosaccharides was performed with $NaBD_4$ in $NH_3$ 2 M, followed by acetylation. The organic phase was washed three times with water and then dried by centrifugal evaporation. The partially methylated alditol acetates were injected and analysed by GC-MS [29].

Identification was achieved by comparing the mass spectra and other spectra with a laboratory made database.

### 2.5. Colour

The retentate water soluble material was added to a commercial tonic water for three different concentrations of 1.0 mg·mL$^{-1}$, 0.25 mg·mL$^{-1}$ and 0.1 mg·mL$^{-1}$. Colour measurements were performed using a CR-400 colorimeter (Konica Minolta, Osaka, Japan) to determine CIE (Commission Internationale de l'Eclairage) tristimulus coordinates: *L\** (lightness), *a\** (red to green) and *b\** (yellow to blue). The samples (4 mL) were placed under the colorimeter beam in a small glass container (2.2 cm of diameter × 1.5 cm of height). Absorbance was read at 520 nm, the wavelength of maximum absorbance. Changes in both parameters were monitored after 22 and 72 days of storage. The pink tonic water was kept in a dry place away from sunlight.

### 2.6. Statistical Analysis

Colour parameters and absorbance results were subjected to three-factor analysis of variance (ANOVA), $p < 0.05$ was considered statistically significant, and followed by a

multiple range test, Tukey's range test. Minitab Statistical Software 20.4.0 for windows (trial version, Minitab, State College, PA, USA).

## 3. Results and Discussion

### 3.1. Concentrate Elderberry Juice Carbohydrates Characterization

The concentrate elderberry juice contains 74% of total sugars and 55% free sugars (Table 1). Considering that the concentrated juice had 65 °Brix, that is, the amount of total soluble solids in the product, these values show that sugars contribute only to 48 °Brix. The difference seems to be due to the presence of anthocyanins, which account for 11.2 to 18.5% on elderberry fruits of the varieties under study, considering three harvest years [1]. Total sugar values are 4–8-fold higher than the ones present in the fruit itself (9–17%), the maximum concentration ratio attained with the process of concentration [1]. Glucose (57%) is the main sugar followed by fructose (40%), which is consistent with the reported composition for elderberry fruit and other types of wild berries [1]. Sucrose was not observed in concentrated juice. Although its presence in low amounts has been described to exist in the fruit, the high temperatures used in juice processing may promote the hydrolysis of this disaccharide [4].

**Table 1.** Sugar composition of elderberry concentrate juice (ECJ), expressed in molar percentage (mol%) and amount of total sugars ($\mu g \cdot mg^{-1}$).

| | η% [1] | (%mol) | | | | | | | | | Total Sugars |
|---|---|---|---|---|---|---|---|---|---|---|---|
| | | Rha | Fuc | Ara | Xyl | Man | Gal | Glc | Fru | UA | ($\mu g \cdot mg^{-1}$) |
| ECJ Free sugars | | 0.3 | 0.5 | 0.4 | 2 | | | 57 | 40 | | 552 ± 0.3 |
| ECJ Total sugars | | 0.2 | 0.5 | 0.3 | 3 | 6 | | 59 | 30 | 1 | 738 ± 0.1 |
| Polymeric material | 1 | 1 | 2 | 5 | 22 | 13 | 2 | 44 | | 11 | 279 ± 21 |
| Hydrophilic material | 0.2 (17) | 2 | 1 | 4 | 5 | 1 | 5 | 6 | | 77 | 687 ± 34 |
| Hydrophobic material | 0.5 (40) | 1 | | 5 | 28 | | | 65 | | 1 | 204 ± 1 |

[1] Values between brackets correspond to yields calculated in polymeric material weight basis. Relative standard deviation of sugars mol% is less than 5% for major sugars.

The high molecular weight material represents 1% of the concentrated juice. This material consists of 28% sugars, mostly glucose (44%), followed by xylose (22%), and similar amounts of mannose (13%) and uronic acids (11%). Reverse solid phase extraction allowed us to separate an aqueous fraction, representing 17% of the high molecular weight material (HMWM), and a methanolic fraction, representing 40% of the HMWM. The hydrophilic material contained 69% of sugars, mainly composed of uronic acids (77%), consistent with the presence of pectic polysaccharides branched through rhamnose (2%) by small chains of galactose (5%) and arabinose (4%). In fact, although commercial pectinolytic enzymes are used during juice processing, not all linkages are hydrolysed, as also described for black currant and bilberry juices [31]. The presence of xylose (5%), as well as rhamnose, can also be associated to pectic polysaccharides or polymerized anthocyanins. Polyphenols can be covalently linked to polysaccharides and also interact non-covalently [25]. Most polymerized anthocyanins interact non-covalently and were recovered in hydrophobic fraction, comprising 20% of sugars (Table 1), mainly glucose (65%) and xylose (28%). These sugars are the carbohydrate moieties of the polymerized anthocyanins containing cyanidin 3-glucoside, cyanidin 3-sambubioside (glucose and xylose disaccharide residues), and cyanidin 3-rutinoside (glucose and rhamnose disaccharide residues), which are the main elderberry anthocyanins [17,31].

### 3.2. By-Products Characterization

The production of concentrated elderberry juice from elderberry must results in 25% of pomace. This by-product is rich in carbohydrates, which account for 50% of total sugars (Table 2). This value is close to the 43–45 g/100 g reported for bilberry pomace [32]. Glucose (50%), xylose (24%), and galacturonic acid (14%) are the major sugar residues

present in elderberry pomace, which are characteristic of plant cell wall polysaccharides and anthocyanins present in elderberry skins, namely cyanidin-3-glucoside and cyanidin-3-sambubioside. Accordingly, bilberry skins have been reported to contain 20-fold higher concentration of anthocyanins when compared with their pulp [32]. Elderberry pomace contains also 15% of free sugars, namely xylose (61%), glucose (19%), and only a residual amount of free fructose (5%). As the major free sugars present in the juice are glucose and fructose, the presence of xylose and the majority of glucose may result from the enzymatic hydrolysis of cyanidin-3-sambubioside after the pressing processing step. The pomace free sugars composition also allows to infer that most of the juice free sugars have been removed.

**Table 2.** Sugar composition and mannitol of elderberry pomace and retentate fractions.

| | η% [1] | (mol%) | | | | | | | | | Total Sugars | Mannitol |
| | | Rha | Fuc | Ara | Xyl | Man | Gal | Glc | Fru | GalA | (μg·mg$^{-1}$) | (μg·mg$^{-1}$) |
|---|---|---|---|---|---|---|---|---|---|---|---|---|
| **Pomace** | | | | | | | | | | | | |
| Total sugars | | 0.5 | 0.2 | 4 | 24 | 4 | 3 | 50 | 1 | 14 | 496 ± 6 | |
| Free sugars | | | | 9 | 61 | | 5 | 19 | 5 | | 152 ± 0.01 | |
| **Retentate** | | | | | | | | | | | | |
| WSM | 48 | | | | | | | | | | | |
| Total sugars | | 2 | 0.1 | 3 | 6 | 11 | 4 | | 47 | 27 | 685 ± 34 | 9 ± 0.03 |
| Free sugars | | 4 | 0.4 | 4 | 5 | | 1 | | 80 | 8 | 466 ± 0.05 | |
| LMWM | 38 (77) | 1 | | 2 | 3 | | 1 | | 79 | 14 | 317 ± 0.2 | 10 ± 0.01 |
| HMWM | 7 (14) | 6 | | 12 | 10 | 4 | 13 | 1 | 4 | 47 | 526 ± 36 | 4 ± 0.01 |
| Hydrophilic material | 1.3 (38) | 8 | 1 | 8 | 4 | 1 | 16 | 3 | | 61 | 719 ± 12 | |
| Hydrophobic material | 1.5 (44) | 3 | | 8 | 23 | | 7 | 44 | | 15 | 201 ± 0.05 | |
| Washing | 0.52 | | | | | | | | | | | |
| Total sugars | | 2 | 0.1 | 2 | 7 | 16 | 2 | | 39 | 31 | 639 ± 11 | 8 ± 0.03 |
| Free sugars | | 5 | 0.2 | 6 | 19 | | 7 | | 63 | | 182 ± 0.02 | |
| WIM | 50 | | | | | | | | | | | |
| Total sugars | | 1 | | 3 | 12 | 9 | 10 | 46 | | 8 | 256 ± 6 | |

[1] Values between brackets correspond to yields calculated in water soluble material or HMWM dry weight basis. WSM—Water soluble material; LMWM—Low molecular weight Material; HMWM—High molecular weight material; WIM—Water insoluble material. Relative standard deviation of sugars mol% is less than 5% for the major sugars.

The retentate is the result of the accumulation of carbohydrates, polyphenols, proteins and lipophilic compounds retained by the ultrafiltration membrane [20], and it accounts for 4.4% of initial must. The centrifugation of this by-product resulted in a supernatant with water soluble material (WSM), representing 48% of the initial mass and containing 69% of sugars. The major sugars in the retentate supernatant are fructose (47%) and galacturonic acid (27%). Free sugars represent 47% of the WSM, being mostly comprised of fructose (80%). There is also mannitol (1%) present in elderberry water soluble material recovered from the retentate. The occurrence of mannitol, as well as other polyols such as xylitol, seems to be a characteristic of wild berries [33]. The low molecular weight material (LMWM) of the retentate, recovered after dialysis, accounted for 77% of the retentate WSM, representing 38% of the total mass of the retentate. Its sugars composition is similar to the one determined for WSM free sugars, showing that the LMSM was composed mainly by free sugars, mainly fructose and 1% mannitol. The remaining material seems to be anthocyanins, consistent with its strong purple colour. The high molecular weight material (HMWM) represents 14% of the WSM, on a dry weight basis, showing that there is 7% soluble material of high molecular weight in the retentate. This material consists of 53% sugars, mostly galacturonic acid (47%), followed by galactose (13%) and arabinose (12%), suggesting the presence of pectic polysaccharides branched by galactose and arabinose side chains [34].

As a result of the reverse phase extraction of the HMWM from the retained material, a hydrophobic fraction which accounted for 44% of the total HMWM mass was obtained.

It contained only 20% of sugars, mainly glucose (65%), xylose (28%), and rhamnose (3%), possibly associated with polymerized anthocyanins containing cyanidin 3-glucoside, cyanidin 3-sambubioside and cyanidin 3-rutinoside monomers. This hydrophobic fraction also contained galacturonic acid, arabinose, and galactose, which could be due to the presence of methyl esterified galacturonic acid, allowing the retention of these uncharged polysaccharides in the C18 column, or arabinan complexes with polyphenols [34]. On the contrary, the hydrophilic fraction, accounting for 38% of total HMWM mass, had 79% of carbohydrates rich in galacturonic acid (61%), consistent with the enrichment of pectic polysaccharides branched by chains of galactose (16%) and arabinose (8%). This was confirmed by the results of a methylation analysis (Table 3).

**Table 3.** Glycosidic-linkage of hydrophilic fraction obtained upon solid-phase extraction of the high molecular weight material from the water-soluble fraction of the retentate.

| Glycosidic-Linkage | | %mol |
|---|---|---|
| t-Rha*p* | | 3.3 |
| 2-Rha*p* | | 5.8 |
| 2,4-Rha*p* | | 3.5 |
| 2,3-Rha*p* | | 1.5 |
| 2,3,4-Rha*p* | | 1.6 |
| | Total Rha*p* | 15.8 (19.5) |
| t-Fuc*p* | | 1.2 |
| | Total Fuc*p* | 1.2 (2.4) |
| t-Ara*f* | | 6.2 |
| 2-Ara*f* | | 0.7 |
| 3-Ara*f* | | 2.7 |
| 5-Ara*f* | | 1.1 |
| 4-Ara*p* | | 1.6 |
| 3,5-Ara*f* | | 0.5 |
| 2,3-Ara*p* | | 6.4 |
| 2,3,5-Ara*f* | | 2.0 |
| | Total Ara | 21.3 (19.5) |
| t-Xyl*p* | | 3.7 |
| 2-Xyl*p* | | 0.5 |
| 4-Xyl*p* | | 6.2 |
| 2,4-Xyl*p* | | 2.9 |
| | Total Xyl*p* | 13.2 (9.8) |
| t-Man*p* | | 0.3 |
| 3,4-Man*p* | | 1.7 |
| 2,6-Man*p* | | 0.6 |
| | Total Man*p* | 2.7 (2.4) |
| t-Gal*p* | | 6.8 |
| 2-Gal*p* | | 0.5 |
| 4-Gal*p* | | 1.0 |
| 3-Gal*p* | | 0.5 |
| 6-Gal*p* | | 4.2 |
| 2,4-Gal*p* | | 2.6 |
| 3,6-Gal*p* | | 7.0 |
| 2,3,4-Gal*p* | | 1.1 |
| 2,3,4,6-Gal*p* | | 1.3 |
| | Total Gal*p* | 25.0 (39.0) |
| t-Gal*p*A | | 2.8 |
| 4-Gal*p*A | | 2.5 |
| 2,4-Gal*p*A | | 1.0 |
| | Total Gal*p*A | 6.3 |
| t-Glc*p*A | | 7.3 |
| 3,4-Glc*p*A | | 2.1 |
| | Total Glc*p*A | 9.3 |

Retentate is rich in 3,6-Gal (7.0%), t-Gal (6.8%), t-Ara*f* (6.2%), 3-Ara*f* (2.7%), 2,3,5-Ara*f* (2.0%), 5-Ara*f* (1.1%), 2-Ara*f* (0.7%), 3,5-Ara*f* (0.5%), characteristic of type II arabinogalactans. These linkages are described in elderberry fruit and apple retentate [6,7,20]. They appear to have side chains of rhamnogalacturonans type II (RG II), as well as arabinans and galactans. Rhamnose linkages, 2-Rha (5.8%) and 2,4-Rha (3.5%), also reinforce the present of RG II, and also 4-GalA, as they compose the backbone of RGII. The percentage of GalA is much lower than determined by alditol acetates, possibly due to the reduction of carboxylic acid low yield [29,35]. Galacturonic acid linkages are also the main structural feature in homogalacturans [36].

The presence of 4-Xyl (6.2%) and 2,4-Xyl (2.9%) are diagnostic of xylans, probably glucuronoxylans due to the presence of t-GlcA (7.3%), although the carboxyl reduction of uronic acids is not quantitative [30]. The determination of 3% 2,4-Xyl indicates that the xylan may be branched, although low branched (<1%) xylans are reported for grape skins [37]. However, it is possible that, in this residue, the action of endogenous elderberry xylanases, as well as the cocktail of enzymes used as adjuvants for juice extraction, result in the enrichment of highly branched not degraded water soluble motifs, as observed by the xylanase action on brewers spent grain arabinoxylans [38]. These treatments are also able to increment the motives containing acetyl groups at in the xylan backbone, which tend to be resistant to the alkali conditions used for methylation analysis [38]. Some xyloglucan-type polysaccharides could also be present in minor amounts since 4,6-Glc (1.5%) and the presence of diagnostic 2-Xyl (0.5%), together with 2-Gal (0.5%), terminally linked Fuc (1.2%), t-Xyl (3.7%), and t-Gal (6.8%) residues. Xylan and xyloglucan fragments could be explained by the high solubility of this branched polysaccharide in the juice, but due to the high molecular weight, it stays retained in the ultrafiltration membranes, also reported for apple juice retentate [20].

Furthermore, terminally linked rhamnose, xylose, and glucose linkages can corroborate the hypothesis of the presence of polymerized anthocyanins containing the monomers of cyanidin 3-glucoside, cyanidin 3-sambubioside and cyanidin 3-rutinoside.

The washing of the residue only allowed to recover 0.5% of the material, with a composition very similar to the WSM, but only with 18% of free sugars, 1% mannitol, but with higher content of polysaccharides (46%). This sugars composition allows to pool this fraction together with the WSM one.

The insoluble residue (WIM) recovered after the water extraction represented 50% of the starting material, comprising only 26% of carbohydrates, mostly glucose (46%), followed by xylose (12%), galactose (10%) and galacturonic acid (8%), suggesting the presence of glucans and pectic polysaccharides. Glucose should arise from cellulose and xyloglucans present in the cell walls, which, due to the high molecular weight and/or high hydrodynamic volume, were retained in the industrial ultrafiltration membranes [20]. The WIM had also a strong purple colour, which indicates the presence of highly polymerized anthocyanins, eventually covalently linked to pectic polysaccharides.

### 3.3. Retentate Application as a Water-Soluble Colour Ingredient

The water-soluble material (WSM), containing pectic polysaccharides and anthocyanins, and the hydrophobic fraction obtained by reverse phase extraction of the HMWM, composed mainly by anthocyanins, were added to a commercial tonic water, an alcohol-free carbonated drink. Different final concentrations allowed to obtain purple to pink grade scale colour options (Figure 2) and absorbance intensities with a maximum at 520 nm (Table 4), which are characteristic of the anthocyanins. The extracts added maintained the characteristic bitter and sweet flavour of the tonic water. Although the WSM fraction was rich in fructose (Table 2), the sweetness of the tonic water did not increase, probably explained by the high amount of sugar already present in the beverage (8.4 g in 100 mL).

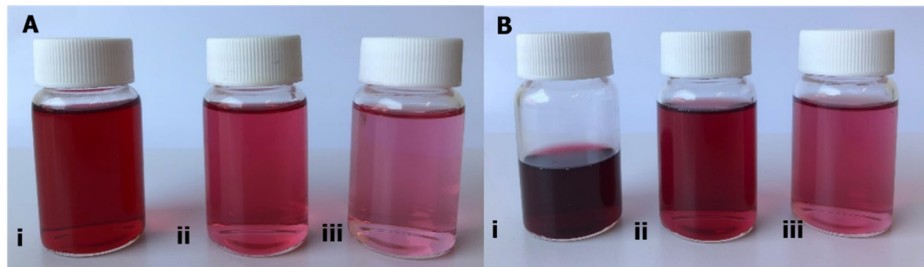

**Figure 2.** Tonic water with addiction of (**A**) water soluble material (WSM) from the retentate or (**B**) hydrophobic material, obtained by solid-phase extraction from the WSM, as a natural colourant, (i) 1 mg·mL$^{-1}$, (ii) 0.25 mg·mL$^{-1}$, (iii) 0.1 mg·mL$^{-1}$.

The use of isolated anthocyanins (Figure 2B) resulted in a tonic water with lower *L*\*, *a*\* and *b*\* values (Table 5), corresponding to a more purple bluish red tone, when compared to the colour of the beverage when the WSM fraction was added (Figure 2A). As the hydrophobic material obtained from solid-phase extraction was only 6.16% of the WSM, for the same final concentration in the beverage, the isolated anthocyanins were 16 times more concentrated, explaining the higher colour intensity, reflected in an increment in the absorbance values at 520 nm of almost three times. The colour variation is only due to the sample concentration since the pH of the tonic water was 3 and did not change with extract addition. However, the colour of elderberry anthocyanins is pH-dependent for neutral and basic pH (Supplementary Material Figure S1 and Table S1). The pink colour is maintained from pH 3 until pH 6. The anthocyanins shift from a flavylium cation at pH 3–6 to a neutral quinoidal base at pH 7, which causes the colour to shift from red to purple. At pH 7, an equilibrium between the blue and the pink (*a*\* and *b*\* values) was observed (Table S1). This is in accordance with the maximum absorbance at 520 nm of the sample at pH 3 and the maximum absorbance at 580 nm at pH 8 (Figure S1).

The comparison of the CIELAB parameters between day 0 and day 22 for the hydrophobic material showed a statical difference in the *b*\* parameter increase, meaning that the tonic water colour changed its colour from an initial shade of bluish and purple red towards an orange tone, although visually imperceptible. No significant differences were observed between day 22 and 72. On the contrary, the addition of WSM did not change the colour at day 22, and at day 72, only slight variations were seen (Table 5), in accordance with the decrease of the absorbance at 520 nm (Table 4). This may be explained by the high instability and susceptibility to degradation of isolated anthocyanins, leading to colour fading and conditioning shelf life and product acceptability, and limiting their industrial application [39,40]. Colour lastingness when using the entire soluble fraction could be explained by a chemical stabilisation of anthocyanins, forming complexes with pectin highly present in this retentate fraction through electrostatic interactions, hydrogen bonding and hydrophobic effects, as well as by the acidic pH of the solution, as citric acid is one of the ingredients of the original tonic water [41,42]. These results showed that it is more advantageous to use the total water-soluble material as a colourant, retarding the fading process and maintaining the required colour for a longer period.

**Table 4.** Absorbance at 520 nm of the tonic water and retentate water soluble material solution, at different concentrations, in the initial day and after 22 and 72 days of storage.

| Tonic Water Formulations | Day 0 | | Day 22 | | Day 72 | |
|---|---|---|---|---|---|---|
| | WSM | hydrophobic | WSM | hydrophobic | WSM | hydrophobic |
| 1.0 mg·mL$^{-1}$ | 1.40 ± 0.02$^{aaa}$ | 3.81 ± 0.02$^{baa}$ | 1.39 ± 0.01$^{aaa}$ | 2.01 ± 0.02$^{bab}$ | 0.95 ± 0.00$^{aab}$ | 2.50 ± 0.08$^{bac}$ |
| 0.25 mg·mL$^{-1}$ | 0.46 ± 0.01$^{aba}$ | 1.17 ± 0.00$^{bba}$ | 0.36 ± 0.01$^{abb}$ | 0.61 ± 0.01$^{bbb}$ | 0.22 ± 0.00$^{abc}$ | 0.87 ± 0.15$^{bbb}$ |
| 0.10 mg·mL$^{-1}$ | 0.19 ± 0.00$^{aca}$ | 0.51 ± 0.03$^{bca}$ | 0.17 ± 0.00$^{acb}$ | 0.24 ± 0.00$^{bcb}$ | 0.08 ± 0.00$^{acc}$ | 0.25 ± 0.00$^{bcb}$ |

[1] Each value is represented as mean ± standard deviation ($n = 3$); different letters indicate significant difference ($p < 0.05$), being the first letter regarding to colourants, second one between concentrations and the last between days.

**Table 5.** Cielab parameters (*L\**, *a\**, *b\**) of pink coloured tonic water with different concentration of elderberry WSM and hydrophobic material at Day 0 and after 22 and 72 days of storage.

| Tonic Water Formulations | *L\** | | | | | | *a\** | | | | | | *b\** | | | | | |
|---|---|---|---|---|---|---|---|---|---|---|---|---|---|---|---|---|---|---|
| | Day 0 | | Day 22 | | Day 72 | | Day 0 | | Day 22 | | Day 72 | | Day 0 | | Day 22 | | Day 72 | |
| | WSM | hydrophobic | WSM | hydrophobic | WSM | hydrophobic | WSM | hydrophobic | WSM | hydrophobic | WSM | hydrophobic | WSM | hydrophobic | WSM | hydrophobic | WSM | hydrophobic |
| 1.0 mg·mL$^{-1}$ | 43.9 ± 0.3$^{aaa}$ | 35.5 ± 0.4$^{baa}$ | 43.8 ± 0.7$^{aaa}$ | 35.9 ± 0.2$^{baa}$ | 32.7 ± 0.9$^{aab}$ | 38.0 ± 0.7$^{bab}$ | 20.6 ± 0.4$^{aaa}$ | 12.4 ± 1.6$^{baa}$ | 20.7 ± 0.7$^{aaa}$ | 13.7 ± 0.6$^{baa}$ | 26.4 ± 1.2$^{aab}$ | 13.8 ± 0.3$^{baa}$ | 3.3 ± 0.5$^{aaa}$ | 1.7 ± 0.2$^{baa}$ | 3.7 ± 0.3$^{aaa}$ | 2.4 ± 0.2$^{bab}$ | 6.5 ± 0.7$^{aab}$ | 3.6 ± 0.2$^{bac}$ |
| 0.25 mg·mL$^{-1}$ | 48.7 ± 0.5$^{aba}$ | 38.3 ± 0.7$^{bba}$ | 53.8 ± 0.4$^{aba}$ | 40.1 ± 0.4$^{bbb}$ | 42.4 ± 0.3$^{abb}$ | 40.9 ± 0.1$^{bbb}$ | 10.8 ± 0.5$^{aba}$ | 17.5 ± 0.8$^{bba}$ | 9.9 ± 0.2$^{aba}$ | 16.1 ± 0.4$^{bba}$ | 14.0 ± 0.2$^{aba}$ | 11.7 ± 0.8$^{bbc}$ | −0.7 ± 0.4$^{aba}$ | 2.0 ± 0.2$^{baa}$ | −0.6 ± 0.1$^{aba}$ | 2.9 ± 0.2$^{bbb}$ | 1.8 ± 0.1$^{abb}$ | 3.2 ± 0.4$^{bab}$ |
| 0.10 mg·mL$^{-1}$ | 56.3 ± 0.4$^{aba}$ | 43.6 ± 0.3$^{bca}$ | 53.0 ± 1.5$^{abb}$ | 46.5 ± 0.3$^{bca}$ | 49.7 ± 0.2$^{acc}$ | 45.9 ± 0.8$^{bca}$ | 5.0 ± 0.1$^{aca}$ | 17.5 ± 0.4$^{bba}$ | 4.4 ± 0.2$^{acb}$ | 13.1 ± 0.3$^{baa}$ | 7.1 ± 0.3$^{acc}$ | 8.0 ± 0.2$^{bca}$ | −1.5 ± 0.0$^{aca}$ | 0.9 ± 0.1$^{bba}$ | −1.1 ± 0.2$^{aca}$ | 1.9 ± 0.0$^{bcb}$ | 0.3 ± 0.1$^{acc}$ | 2.5 ± 0.1$^{bbb}$ |

[1] Each value is represented as mean ± standard deviation ($n = 5$); for each colour parameter, different letters indicate significant differences ($p < 0.05$), being the first letter regarding to colourants, the second one between concentrations and the last between days.

## 4. Conclusions

The concentrate elderberry juice contains 61% of sugars, which is slightly lower than the 65 °Brix value overestimated by the presence of anthocyanins. Free sugars represent 55% of the juice, mainly glucose and fructose, also containing pectic polysaccharides. However, a large part of polysaccharides remained in by-products able to be further explored. Pomace contained 50% of total sugars, where glucose was the major residue present (50%), arising from cellulose and xyloglucans, alongside xylose (24%), which is also a moiety of the main anthocyanins present in elderberry. Retentate, resultant from ultrafiltration process, is also rich in carbohydrates (46%), mostly pectic polysaccharides and polymerized anthocyanins, but also fructose (18%) and mannitol (0.9%). Therefore, it was valorised as a natural colourant to produce a pink tonic water with stable colour for over more than 22 days, longer when compared with the isolated anthocyanins. There is also insoluble polymeric material (50%), from which 46% is glucose, probably arising from cellulose, xyloglucans and the glycosyl moiety of the elderberry polymerized anthocyanins that has potential to be used for dietary fibre products.

**Supplementary Materials:** The following are available online at https://www.mdpi.com/article/10.3390/app12199463/s1, Figure S1: Spectra of the absorbance of elderberry juice in citrate and phosphate buffers solutions in the pH range from 3 to 8 (0.2 mL of sample was added to 3.8 mL of each buffer solution), Table S1: Colour values (*a\** and *b\**) of elderberry juice in buffer solutions in the pH range from 3 to 8.

**Author Contributions:** Conceptualization, O.T., E.C. and M.A.C.; methodology, E.C. and M.I.V.; validation, E.C. and M.A.C.; formal analysis, M.I.V.; investigation, M.I.V., E.C. and M.A.C.; resources, M.A.C.; data curation, M.I.V., E.C. and M.A.C.; writing—original draft preparation, M.I.V.; writing—review and editing, E.C. and M.A.C.; visualization, E.C. and M.A.C.; supervision, E.C. and M.A.C.; project administration, O.T., E.C. and M.A.C.; funding acquisition, O.T., E.C. and M.A.C. All authors have read and agreed to the published version of the manuscript.

**Funding:** This research was funded by Project BagaConValor—Criação de valor no processo tecnológico de produção de sumo concentrado de baga de sabugueiro, Project n. 033558 (POCI-01-0247-FEDER-033558), and FCT/MEC financial support for LAQV/REQUIMTE (UIDB/50006/2020 & UIDP/50006/2020), through national funds, and the co-funding by the FEDER, within the PT2020 Partnership Agreement and Compete 2020. The E.C. research contract (CDL-CTTRI-88-ARH/2018—REF. 049-88-ARH/2018) was funded by national funds (OE), through FCT, in the scope of the framework contract foreseen in the numbers 4, 5 and 6 of the article 23, of the Decree-Law 57/2016, of 29 August, changed by Law 57/2017, of 19 July.

**Institutional Review Board Statement:** Not applicable.

**Acknowledgments:** Authors thank to Andreia Ferreira for the determination of colour and absorbance of elderberry juice at different pH values.

**Conflicts of Interest:** The authors declare no conflict of interest.

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
