# Peer review of "Elderberry Concentrate Juice Industrial By-Products Characterization and Valorisation"

_applsci, doi:10.3390/app12199463_

Round 1

Reviewer 1 Report (Previous Reviewer 2)

Readers would be interested in your data of anthocyanins from those samples instead of previous publications, and also to observe the anthocyanin content changes during the 22 days. But fine, I would agree with the paper published in the journal.

Author Response

Answers to the Round 1 Reviewer comments

Manuscript ID: applsci-1924588

Elderberry concentrate juice industrial by-products characterization and valorisation

Maria Inês Veloso, Elisabete Coelho, Oswaldo Trabulo, and Manuel A. Coimbra

-Reviewer #1

Comment 1:

“Readers would be interested in your data of anthocyanins from those samples instead of previous publications, and also to observe the anthocyanin content changes during the 22 days. But fine, I would agree with the paper published in the journal.”

Authors answer 1:

    The authors thank the reviewer comment and suggestions to improve the manuscript. However, as stated before this study is focused on the analysis and characterization of the polysaccharides present, allowing to observe that they make the colour of anthocyanins more stable.

Reviewer 2 Report (New Reviewer)

1. It is an interesting article, but I consider and recommend that it be improved and supported by more analyzes both for the raw material and for the finished product.

2. In the introduction, the therapeutic effects of Elderberry (Sambucus nigra L.) (1-4) are mentioned, therefore I recommend other analyzes for tonic water, in order to highlight the biologically active compounds from elderberry fruits that are found in tonic water . Of interest are the phenolic compounds, the antioxidant capacity (DPPH, ABTS, FRAP) and the anthocyanins, only the analyzes performed for the finished product are not sufficient from my point of view.

Author Response

Answers to the Round 1 Reviewer comments

Manuscript ID: applsci-1924588

Elderberry concentrate juice industrial by-products characterization and valorisation

Maria Inês Veloso, Elisabete Coelho, Oswaldo Trabulo, and Manuel A. Coimbra

-Reviewer #2

Comment 1:

“1. It is an interesting article, but I consider and recommend that it be improved and supported by more analyzes both for the raw material and for the finished product.”

Authors answer 1:

    The authors thank the reviewer comment and recommendation to improve the manuscript. However, as stated before this study is focused on the analysis and characterization of the polysaccharides present, allowing to observe that they make the colour of anthocyanins more stable. In terms of free sugars and polysaccharides the raw materials were extensively characterized as shown in table 1, 2 and 3.

Comment 2:

“2. In the introduction, the therapeutic effects of Elderberry (Sambucus nigra L.) (1-4) are mentioned, therefore I recommend other analyzes for tonic water, in order to highlight the biologically active compounds from elderberry fruits that are found in tonic water . Of interest are the phenolic compounds, the antioxidant capacity (DPPH, ABTS, FRAP) and the anthocyanins, only the analyzes performed for the finished product are not sufficient from my point of view.”

Authors answer 2:

The authors with this study intend to highlight the potentiality of the by-products obtained during concentrate elderberry juice production, namely the retentate. The application of the extracts obtained from retentate were studied showing the importance of the presence of pectic polysaccharides in the colour stability. Showing that the simple recovery of a soluble fraction from the retentate is more efficient in terms of colour stability than the isolated pure anthocyanins from the retentate. Moreover, the biological active compounds and related properties were mentioned in the introduction as state of the art not as aim of the work. The tonic water final product has the purpose to be pink due to the addition of anthocyanins. The manuscript showed that anthocyanins with a naturally stabilizer agent, the pectic polysaccharides maintained the pink colour for longer time.

Reviewer 3 Report (New Reviewer)

Dear authors,

After the review process, I have several comments: in the abstract, you should clearly mention the aim of the paper, not only data; you should include more numerical data; you should mention how figure 1 was realized - copyright condition?!; the discussion of the results are poor, you should add new comments based on the bioactive potential of functional products and bioavailability; limitation of the study should be clearly mentioned and future valorization based on the correlation between microbiota bioactivity and bioavailability of functional compounds; the paper should be corrected for minor language errors.

Best regards!

Author Response

Answers to the Round 1 Reviewer comments

Manuscript ID: applsci-1924588

Elderberry concentrate juice industrial by-products characterization and valorisation

Maria Inês Veloso, Elisabete Coelho, Oswaldo Trabulo, and Manuel A. Coimbra

-Reviewer #3

Comment 1:

“After the review process, I have several comments: in the abstract, you should clearly mention the aim of the paper, not only data; you should include more numerical data;”

Authors answer 1:

The authors thank the reviewer comment and suggestions to improve the manuscript. The abstract was improved. A sentence with the aim of the work was included as well as the numerical data supporting the more stability attained by the complex pectic polysaccharides-anthocyanins in comparison to the isolated anthocyanins.

Comment 2:

“; you should mention how figure 1 was realized - copyright condition?!;”

Authors answer 2:

The figure 1 is the flowchart used in the fractionation of the elderberry juice retentate. This is not copyright condition because it is an original figure. This procedure was adapted from apple juice retentate fractionation. Only the first centrifugation and the wash of the residue were similar to the apple retentate procedure previously reported. A completely different strategy was used in the present study, it was performed the isolation of the compounds present in the water soluble material.

Comment 3:

“; the discussion of the results are poor, you should add new comments based on the bioactive potential of functional products and bioavailability;”

Authors answer 3:

The discussion is in line with the results presented in the manuscript that follows the aim and objective of the present work. This study is focused on the analysis and characterization of the polysaccharides present, allowing to observe that they make the colour of anthocyanins more stable. This property was explored to produce a pink ingredient to be used in tonic water. The bioactive potential and bioavailability of the anthocyanins and polysaccharides present is not the scope of the present work. The colour properties and its stability along time is an objective that is presented and discussed.

Comment 4:

“limitation of the study should be clearly mentioned and future valorization based on the correlation between microbiota bioactivity and bioavailability of functional compounds;”

Authors answer 4:

The valorisation of pomace and retentate is based on their properties as a colourant agent, their potentiality as a technological ingredient is discussed in the present work. The valorisation of these by-products is never focused on the production of a functional ingredient only as a colouring one. The correlation between microbiota bioactivity and bioavailability of functional compounds is out of the scope of the present work.

Comment 5:

“the paper should be corrected for minor language errors.”

Authors answer 5:

The paper was revised, and some language errors were corrected.

Round 2

Reviewer 3 Report (New Reviewer)

No other comments

This manuscript is a resubmission of an earlier submission. The following is a list of the peer review reports and author responses from that submission.

Round 1

Reviewer 1 Report

Its a great study addressing the possibilities for increasing the value of elderberry juice industry by products. It is well designed and well written. 

I see that this study used the byproducts of three cultivars “Bastardeira”, “Sabugueira” and “Sabugueiro”, would the results and composition be different if individual cultivars were collected separately?

Figure 2 should be formatted better

Reviewer 2 Report

The research is interesting, but critical parameters have yet to be measured, such as pH,  total anthocyanidin and total phenolic content. Better have individual phenolics and anthocyanidin analysis by HPLC-DAD-MS or QToF-MS.

Line 82. Please provide the protocol and membrane material, pressures etc. of ultrafiltration membranes of 100 kDa.

Line 90. Why not use acidified water?

Line 135. What were the injection volume and split ratio?

Line 166. What was the container used for color measurement?

Line 178. Please measure the total anthocyanidin and total phenolic content of each sample.

Line 212. please provide standard deviations for all values in Table 1. 

Line 231. please provide standard deviations for all values in Table 2. 

Line 329. What was the pH of each bottle? Color is pH-dependent.

Reviewer 3 Report

This article shows characterization of elderberry byproduct that does not imply any novelty in terms of extraction or characterization methods. Furthermore, the valorsitation consists in the formulation of tonic water. In order to be considered a potential ingredient (even if a natural coloring) some kind of sensory analyses or consumer perception or benchmarking would add value.

Furthermore, there are approximately 11 self-citations and that arise some ethical concerns.

In its present form, the article lacks novelty and interest to the readers.

Reviewer 4 Report

This study was well designed and conducted in a comprehensive manner, I suggest accepting the paper.